# PM-YOLO: A Powdery Mildew Automatic Grading Detection Model for Rubber Tree

**DOI:** 10.3390/insects15120937

**Published:** 2024-11-28

**Authors:** Yuheng Li, Qian Chen, Jiazheng Zhu, Zengping Li, Meng Wang, Yu Zhang

**Affiliations:** 1School of Cyberspace Security (School of Cryptology), Hainan University, Haikou 570228, China; liyuheng@hainanu.edu.cn; 2Key Laboratory of Internet Information Retrieval of Hainan Province, Haikou 570228, China; 3School of Tropical Agriculture and Forestry, Hainan University, Danzhou 571737, China; zhujiazheng2022@163.com (J.Z.); lzping301155@126.com (Z.L.); wangmeng@hainanu.edu.cn (M.W.); 4Sanya Institute of Breeding and Multiplication, Hainan University, Sanya 572025, China

**Keywords:** rubber tree, powdery mildew, deep learning, automatic grade, real-time detection

## Abstract

Powdery mildew is a significant disease affecting rubber trees, which reduces yield and quality by forming white fungal patches on leaves. Although in recent years, object detection technologies in agriculture application have improved, many cases especially small-scale regions remain undetected. In this study, we proposed PM-YOLO, an advanced detection model for automatically grading rubber tree powdery mildew. The proposed model integrates innovative modules to achieve superior performance, (1) Feature Focus and Diffusion Mechanism (FFDM) enhances multi-scale feature integration, and (2) Dimension-Aware Selective Integration (DASI) module optimizes the detection of small targets. Additionally, we developed an automatic grading algorithm to quantify disease severity using a precise, formula-driven approach based on leaf damage. Furthermore, we constructed a powdery mildew datasets containing 6200 images with 38,000 annotations for powdery mildew detection task. Experimental results demonstrated that PM-YOLO outperformed the state-of-the-art methods in precision and recall. This work offers an approach for early detection and management of powdery mildew, thereby promoting sustainable rubber tree cultivation.

## 1. Introduction

Rubber tree (*Hevea brasiliensis* Muell. Arg.), a vital economic crop and the primary source of natural rubber, is highly susceptible to powdery mildew (PM), caused by *Erysiphe quercicola* S. Takam. & U. Braun. This disease predominantly affects young tissues, manifesting as white, powdery spots on leaves, stems, and buds [1]. As the infection progresses, these spots expand and coalesce, leading to yellowing, curling, and premature defoliation, significantly reducing the photosynthetic capacity and natural rubber production. Thriving in hot, humid conditions, *E. quercicola* spreads rapidly in densely planted rubber forests through airborne conidia, which germinate under favorable conditions. The pathogen’s life cycle includes both asexual reproduction for rapid spread and sexual reproduction for survival in adverse environments. Without early intervention, the disease’s rapid proliferation severely hinders tree growth and productivity, highlighting the need for timely detection and effective management strategies.

The traditional methods for detecting powdery mildew in rubber trees include symptom observation, microscopic examination, and molecular biological techniques [2]. Symptom observation involves the initial appearance of white powdery substances on the leaves, while microscopic examination confirms the presence of the pathogen. Additionally, molecular techniques such as PCR and LAMP use specific primers to amplify the DNA of the oidium heveae [3], enabling accurate detection. In recent years, with advancements in remote sensing technology and deep learning, UAV-based remote sensing has shown significant potential in agricultural disease detection. Zeng et al. [4,5] conducted studies on the early detection of rubber tree powdery mildew, focusing on hyperspectral imaging and disease identification at different spatial resolutions, offering new methods for precise monitoring.

Object detection is a crucial task in computer vision, with applications in fields such as autonomous driving, surveillance, and image analysis. Over the years, researchers have developed various methods to enhance the accuracy and efficiency of object detection systems. The integration of deep learning technology has significantly improved both the accuracy and real-time performance of object detection in recent years. Currently, deep-learning-based detection methods are primarily categorized into two types: two-stage detection algorithms and one-stage detection algorithms.

Two-stage detection algorithms, such as R-CNN [6], Fast R-CNN [7], and Faster R-CNN [8], perform object detection in two steps: first, generating candidate regions, and, second, classifying these regions. These algorithms are known for their high recognition accuracy and low false negative rates. However, their detection speed is limited, making them unsuitable for real-time processing scenarios. In contrast, one-stage detection algorithms, including You Only Look Once (YOLO) [9], Single Shot Multibox Detector (SSD) [10], and RetinaNet [11], directly predict the class probabilities and bounding box coordinates of objects. By eliminating the candidate region generation step, these algorithms streamline the detection process, significantly enhancing the detection speed. As a result, they are able to classify and localize objects with low latency, making them more suitable for real-time applications.

Among these methods, YOLO is a deep learning algorithm that is widely used for object detection tasks, known for its efficiency, speed, and accuracy. Its architecture typically includes a backbone network, neck connection layers, and a detection head. From YOLOv1 to YOLOv3, this structure became the foundation of the standard detection framework [9,12,13]. YOLOv4 [14] and YOLOv5 [15] further evolved by introducing CSPNet, replacing DarkNet, and incorporating data augmentation, an improved PAN (Pyramid Attention Network), and optimizations for various model sizes. YOLOv6 [16] optimized both the connection layers and backbone using BiC and SimCSPSPPF, alongside anchor-assisted training and self-distillation strategies. YOLOv7 [17] introduced E-ELAN to enhance gradient flow paths and experimented with various trainable bag-of-freebies techniques. YOLOv8 [18] proposed the C2f module to improve the efficiency of feature extraction and fusion. YOLOv9 [19] improved the network architecture using GELAN and introduced PGI to optimize the training process. YOLOv10 [20] introduced significant improvements in the real-time object detection performance through advanced feature extraction and anchor-free design, enhancing accuracy and efficiency.

Based on the success of deep learning in object detection, many researchers have applied it to crop disease detection with remarkable results. Deep learning has been widely adopted in agricultural detection, outperforming the traditional machine learning methods. Zhang et al. [21] improved Faster R-CNN for detecting diseased soybean leaves by selecting ResNet50 as the backbone network, demonstrating that the enhanced algorithm outperformed YOLOv2, achieving a mean average precision (mAP) of 83.34%. Dananjayan et al. [22] employed multiple models to detect citrus leaf diseases, showing that the Scaled-YOLOv4 model was highly effective for early-stage detection. Agarwal et al. [23] introduced a novel Conv2D model with a modified ReLU activation function for detecting eight types of cucumber leaf diseases, achieving an accuracy of 93.75%. Wang et al. [24] proposed an automatic pest detection method incorporating multi-scale environmental awareness into information representation, which successfully identified pests in real-world production environments. This method overcame the challenges associated with small target size and the complexity of pest identification, yielding strong results. However, these studies focused on targets with well-defined boundaries, whereas, in early-stage crop management, the boundaries are often less distinct. Moreover, environmental factors such as lighting and occlusion pose significant challenges to crop disease detection, indicating that several technical issues remain unresolved.

However, the features of rubber tree powdery mildew are difficult to capture due to the irregularity in size, shape, and distribution, as well as the diversity in the disease development, which leads to limitations in the performance of general object detectors. Furthermore, powdery mildew characteristics evolve rapidly, influenced by factors such as the rubber tree phenological stages, meteorological conditions, and pathogenicity levels. These factors add complexity to the detection, impeding the achievement of high-precision results. To address the aforementioned issues, the main aims of this study were as follows:(1)A new detection framework PM-YOLO for detecting powdery mildew was proposed, which incorporates a Feature Focus and Diffusion Module (FFDM) based on multi-scale fusion.(2)The Dimension-Aware Selective Integration (DASI) module was introduced to enhance the detection of tiny powdery mildew targets.(3)We proposed a deep-learning-based automatic grading algorithm for powdery mildew, achieving precise grading based on the affected area of the leaves.(4)We constructed a new rubber tree powdery mildew dataset, encompassing rubber tree leaves from various types and environments.

## 2. Materials and Methods

### 2.1. Image Datasets

#### 2.1.1. Image Acquisition

Image collection was conducted from March 2023 to April 2024 between 9:00 AM and 12:00 PM and 2:00 PM and 5:00 PM at three natural agricultural parks and a seedling nursery in Hainan Province, China. The agricultural parks include Baoting Li and Miao Autonomous County (109.36° E, 18.32° N), Qiongzhong Li and Miao Autonomous County (109.44° E, 19.21° N), and Xi Liu Avenue Agricultural Park in Danzhou City (109.65° E, 19.57° N), all offering ideal natural field environments for rubber tree growth. The seedling nursery is located at the Seedling Laboratory of the Sanya Institute of Breeding and Multiplication, Hainan University, in Sanya City, Hainan Province (as shown in Figure 1a). This indoor facility is equipped with air conditioning, smart humidifiers, temperature and humidity sensors, and artificial lighting. The main rubber tree varieties photographed were 73,397 and Dafeng 95. The image collection equipment included a Canon 6D Mark II, Canon EOS 90D, Nikon D90, and iPhone 13, capturing images in three resolutions: 4640 × 6960 pixels, 3120 × 3120 pixels, and 6240 × 4160 pixels. Over 9000 images of rubber tree leaves, exhibiting different disease levels, growth forms, and complex backgrounds, were collected. After manual screening and preprocessing, 6200 high-quality images of rubber tree leaves infected with powdery mildew at six disease levels (0, 1, 3, 5, 7, and 9) were curated for subsequent research.

Furthermore, to support ongoing research and disease monitoring, a 4G-enabled high-definition camera (Hikvision DS-NVR-F104/P/4G) was installed at the monitoring site, Jinjiang Farm in Xiangshui Town, Baoting Li and Miao Autonomous County, Hainan Province (as shown in Figure 1b). This camera facilitates remote monitoring and is equipped with a pan–tilt–zoom (PTZ) system that allows preset points to be configured. The camera automatically rotates and adjusts focus at preset intervals every half-day, capturing images from different angles during its patrol. The captured images are transmitted to the front end in real time, enabling monitoring personnel to perform timely detection and disease prevention.

#### 2.1.2. Image Annotation and Analysis

Rubber tree powdery mildew images were manually annotated by experts using the LabelImage rectangular annotation tool to mark all affected areas on the leaves, followed by a grading process. These annotations were then uniformly converted to the YOLO standard dataset format for subsequent training, validation, and testing. In total, 6200 images of rubber tree powdery mildew, containing 38,000 annotated frames, were collected and split into training, validation, and testing sets in an 8:1:1 ratio, resulting in 4960, 620, and 620 images, respectively.

The dataset comprised images of various disease severity levels under different environmental conditions (as shown in Figure 2). The characteristics of powdery mildew exhibited significant variability in size, shape, and distribution, posing challenges for accurate detection. This difficulty was particularly evident in the smaller infected regions, especially under conditions of poor camera focus or heavy leaf occlusion, further complicating the extraction of distinguishing features. To address these challenges, data augmentation techniques such as multi-scale detection, Gaussian noise addition, brightness adjustment, and image rotation were employed, which significantly enhanced the diversity of the training set by generating images under various poses and scenarios.

### 2.2. YOLOv10

YOLOv10 was introduced as a breakthrough in the ongoing development of the YOLO series, focusing on further improving both model efficiency and accuracy. One of its key innovations is the dual-label assignment strategy, which addresses the traditional reliance on Non-Maximum Suppression (NMS) in earlier YOLO algorithms. During training, YOLOv10 utilized both “one-to-one” and “one-to-many” assignment approaches, optimized by a consistent matching metric, leading to a substantial improvement in the prediction head’s output quality. This advancement enabled true end-to-end detection, representing a significant milestone in the field of object detection, as illustrated in Figure 3.

In the two-label assignment process, the matching measure was used to calculate the matching degree between the prediction box and the real box. The matching measure was composed of the classification score and IoU, as shown in Equation (Equation 1).
(1)$$m\left(\alpha ,\beta \right)=s\cdot p^{\alpha }\cdot IoU{\left(\hat{b},b\right)}^{\beta }$$
where *s* represents the spatial prior of the prediction, *p* is the classification score, and $\hat{b}$ and *b* are the coordinates of the predicted bounding box and the ground truth bounding box, respectively. α and β are adjustment parameters that control the impact of semantic classification and localization regression. This matching metric is used for two assignment branches to maintain consistency in the supervision signals.

To ensure the consistency of the optimization direction between the one-to-one head and the one-to-many head, YOLOv10 proposed a consistency matching metric. This was achieved by adjusting the parameters α and β so that both branches shared a consistent optimization goal during training, as shown in Equation (Equation 2). where *r* is a scaling factor that ensures that the matching degree of the one-to-one head is consistent with that of the one-to-many head.
(2)$$m_{o2o}=r\cdot m_{o2m}=r\cdot s\cdot p^{\alpha_{o2m}}\cdot \mathrm{IoU}{\left(\hat{b},b\right)}^{\beta_{o2m}}$$

YOLOv10 optimized other task alignment objectives by measuring the difference between the one-to-many and one-to-one supervision targets using the Wasserstein distance metric, as shown in Equation (Equation 3).
(3)$$A=t_{o2o,i}-I\left(i\in \Omega \right)t_{o2m,i}+\sum_{k\in \Omega \setminus \{i\}}t_{o2m,k}$$
where $t_{o2o,i}$ and $t_{o2m,i}$ represent the classification targets for one-to-one and one-to-many, respectively. I(i∈Ω) is an indicator function, which signifies whether *i* is a positive sample in the one-to-many branch.

YOLOv10 constructs the tenth generation of detection models designed for a variety of application scenarios, including YOLOv10-N, YOLOv10-S, YOLOv10-M, YOLOv10-B, YOLOv10-L, and YOLOv10-X. Among these models, the YOLOv10-N model, due to its smaller model depth and number of parameters, becomes an ideal choice for standard GPU computing. Based on these advantages, we chose YOLOv10-N as the baseline algorithm and customized it to accurately identify rubber tree powdery mildew, meeting the needs of specific feature recognition.

### 2.3. PM-YOLO

#### 2.3.1. PM-YOLO Overall Structure

In this study, we constructed the PM-YOLO detection model based on YOLOv10n (as shown in Figure 4). The key block model of this structure is illustrated in Figure 5. Furthermore, the model incorporates several significant enhancements:

Firstly, we designed a neck framework based on the Feature Focus Diffusion Mechanism (FDM), which effectively diffuses and propagates features across different hierarchical levels. This mechanism enabled better integration of high-level features and low-level features across various scales. Additionally, we developed the Feature Focus Diffusion Model (FFDM), a module built on FDM that selectively focuses on specific layers or features. This approach ensured that the model emphasizes critical and detailed features, enhancing its capability to detect small objects within complex backgrounds. Finally, to address the challenge of detecting small targets related to powdery mildew, we incorporated the Dimension-Aware Selective Integration (DASI) module. This module integrated features across different dimensions, significantly enhancing the detection of small disease areas, particularly for early-stage powdery mildew on rubber trees.

#### 2.3.2. Feature Focus Diffusion Mechanism

In the PM-YOLO model, we proposed FDM as the core architecture, optimizing the neck of the original YOLOv10 to enhance feature propagation across multiple detection scales. FDM effectively diffused the features extracted by the backbone network throughout the model’s layers, enriching contextual information and improving detection accuracy at different scales. Compared to traditional YOLOv10, which relies on single-point connections, FDM enabled more balanced diffusion and integration of high- and low-level features across detection layers, promoting better interaction between them. This design not only enables the model to capture image details more comprehensively but also significantly boosts detection performance at various scales.

In this study, we introduced multi-scale convolution representations to enrich feature extraction and enhance the model’s capture ability through multi-head attention. Subsequently, we performed feature fusion and downsampling to achieve feature focusing. In addition, to enhance feature retention, we introduced a residual connection mechanism that added the input features to the features after convolution. Then, feature weighting was applied based on the different dimensions of the output features, resulting in the final expanded feature representation, which achieved multi-scale feature diffusion. The overall formula is as follows: (4)$$F_{i}^{\left(s\right)}=\sum_{k=1}^{K}\mathrm{Conv}\left(W_{i,k}^{\left(s\right)},X\right)+b_{i}^{\left(s\right)}$$
(5)$$F_{\mathrm{att}}=\sum_{h=1}^{H}\alpha_{h}\left(\mathrm{softmax}\left(\frac{Q_{h}K_{h}^{T}}{\sqrt{d_{k}}}\right)V_{h}\right)$$
(6)$$F_{\mathrm{fuse}}=\sum_{s=1}^{S}\mathrm{Conv}\left(W_{\mathrm{fuse}}^{\left(s\right)},F_{i}^{\left(s\right)}\right)+b_{\mathrm{fuse}}$$
(7)$$F_{\mathrm{final}}=\sum_{d=1}^{D}\beta_{d}\cdot F_{\mathrm{residual}}^{\left(d\right)}=\sum_{d=1}^{D}\beta_{d}\cdot \left(X+F_{\mathrm{down}}^{\left(d\right)}\right)$$
where $F_{i}^{\left(s\right)}$ represents the convolutional features at scale *s*, *K* is the number of convolutional kernels, and $W_{i,k}^{\left(s\right)}$ refers to the weights of the *k*-th kernel. $\alpha_{h}$ denotes the weighting coefficient for the *h*-th attention head, while *r* is the downsampling ratio. Finally, $\beta_{d}$ represents the weighting coefficient for the *d*-th dimension, and $F_{\mathrm{final}}$ is the final multi-scale feature representation.

#### 2.3.3. Feature Focus Diffusion Model

Based on the Feature Focus Diffusion Mechanism, we proposed a multi-scale fusion module, termed FFDM, as illustrated in Figure 6. This module incorporated parallel small-kernel convolutions to capture local information, including ADown and Conv modules, which preserved more information compared to standard convolutions with a stride of 2. Following concatenation through Concat, a set of parallel depthwise convolutions are employed to capture contextual information at multiple scales.

This module accepts inputs at three different scales, ensuring that features at each scale are enriched with contextual information, which enhances subsequent object detection and classification. Additionally, it incorporated a structure similar to the Inception module, capturing rich information across multiple scales through parallel depthwise convolutions. In its design, the module avoided using dilated convolutions to prevent overly sparse feature representations. Instead, it employed 1 × 1 convolutions to fuse local and contextual features, effectively capturing interrelationships between different channels. The 1 × 1 convolution served as a channel fusion mechanism, integrating features from various receptive fields. In this way, the FFDM module not only captured a broad range of contextual information but also preserved the integrity of local texture features.

Additionally, the FFDM introduced the ADown module, an efficient downsampling convolution block specifically designed for object detection tasks. In deep learning models, downsampling is a common technique used to reduce the spatial dimensions of feature maps, enabling the model to capture higher-level image features while decreasing computational load. The core principle of this module can be expressed by the following formula: (8)$$y=x+F\left(x_{1},x_{2},x_{3}\right)$$
where *F* represents the internal processing of the input features within the module, which is calculated by
(9)$$F\left(x_{1},x_{2},x_{3}\right)=\mathrm{PWConv}\left(\mathrm{sum}(\left[x_{1}^{\prime },x_{2}^{\prime },x_{3}^{\prime },\mathrm{DWConvs}\left(x^{\prime }\right)\right])\right)$$
where $x_{1}^{\prime }=\mathrm{UpConv}\left(x_{1}\right)$ applies upsampling and convolution operations on $x_{1}$, $x_{2}^{\prime }=\mathrm{Conv}\left(x_{2},C_{2},\mathrm{hidc}\right)$ if e≠1; otherwise, $x_{2}^{\prime }=x_{2}$, $x_{3}^{\prime }=\mathrm{ADown}\left(x_{3},\mathrm{hidc}\right)$ applies adaptive downsampling and convolution on $x_{3}$. $x^{\prime }=\mathrm{Concat}\left(x_{1}^{\prime },x_{2}^{\prime },x_{3}^{\prime }\right)$ concatenates the processed features along the channel dimension. $\mathrm{DWConvs}(x^{\prime })$ applies a series of depthwise separable convolutions on $x^{\prime }$, while PWConv refers to pointwise convolution operations, used for further feature extraction.

#### 2.3.4. Dimension-Aware Selective Integration Module

The Dimension-Aware Selective Integration (DASI) module is a novel feature fusion strategy designed to address the challenge of integrating high-dimensional and low-dimensional feature information for small-target detection in infrared imagery [25]. During the multi-level downsampling process in object detection, high-dimensional features may lose critical information regarding small targets, while low-dimensional features often lack sufficient contextual information. The DASI module enhanced the saliency of small targets by adaptively selecting and integrating features from different dimensions, ensuring that relevant information from each scale is retained and utilized effectively, as illustrated in Figure 7.

The working principle of the DASI module was is follows. First, the high-dimensional features $F_{h}$ and low-dimensional features $F_{l}$ were aligned with the current layer features $F_{u}$ through convolution and interpolation operations. Then, DASI divided these features along the channel dimension into four parts, resulting in $h_{i}$, $l_{i}$, and $u_{i}$. Each part was merged through an adaptive weight selection mechanism. The calculation formulas were as follows: (10)$${\hat{F}}_{u}=\delta \left(\beta \left(\mathrm{Conv}\left({\left[\alpha u_{i}+\left(1-\alpha \right)h_{i}\right]}_{i=1}^{4}\right)\right)\right)$$
where α is the value obtained by applying the sigmoid activation function to $u_{i}$, and $h_{i}$ represents an input feature map. The convolution operation, Conv(), is applied to a linear combination of the scaled input $au_{i}$ and the transformed $h_{i}$, raised to the fourth power. The parameters β and δ represent batch normalization and a non-linear activation function, respectively, with the final output $\hat{F_{u}}$.

### 2.4. Disease Automatic Grading Algorithm

Currently, the grading of powdery mildew disease primarily relies on manual judgment, which is prone to subjectivity, leading to significant errors and low efficiency. To address this issue, we proposed an automatic grading algorithm for powdery mildew based on object detection, which graded the disease by calculating the area it occupies on the leaf. This algorithm was integrated into PM-YOLO for end-to-end grading detection, as shown in Figure 8.

The algorithm first classified the disease severity according to the industry standard for rubber tree disease grading (as shown in Equation (Equation 14)), dividing powdery mildew into five levels of severity, 1, 3, 5, 7, and 9, based on the affected leaf area percentage.
(11)$$\mathrm{Grade}=\left\{\begin{array}{cc} \mathrm{None}, & \mathrm{if}\;\mathrm{percentage}=0,\\ \mathrm{Level}\_1, & \mathrm{if}\;0<\mathrm{percentage}<1/20,\\ \mathrm{Level}\_3, & \mathrm{if}\;1/20\leq \mathrm{percentage}<1/16,\\ \mathrm{Level}\_5, & \mathrm{if}\;1/16\leq \mathrm{percentage}<1/8,\\ \mathrm{Level}\_7, & \mathrm{if}\;1/8\leq \mathrm{percentage}<1/4,\\ \mathrm{Level}\_9, & \mathrm{if}\;\mathrm{percentage}\geq 1/4. \end{array}\right.$$

To calculate the Intersection over Union (IoU) between two bounding boxes, the algorithm first determined the overlapping region by calculating the $x_{\mathrm{left}}$ and $y_{\mathrm{top}}$ coordinates of the intersection and the $x_{\mathrm{right}}$ and $y_{\mathrm{bottom}}$ coordinates. If there is no overlap, the IoU is 0. Otherwise, the intersection area was computed as the product of the width and height of the overlapping region. The IoU was then calculated as the ratio of the intersection area to the union of both bounding box areas, excluding the intersection area. The calculation formulas were as follows: (12)$$A_{I}=\left(x_{\mathrm{right}}-x_{\mathrm{left}}\right)\times \left(y_{\mathrm{bottom}}-y_{\mathrm{top}}\right)$$
(13)$$A_{i}=\left(x_{\mathrm{right}}-x_{\mathrm{left}}\right)\times \left(y_{\mathrm{bottom}}-y_{\mathrm{top}}\right)$$
(14)$$\mathrm{IoU}=\frac{A_{I}}{A_{1}+A_{2}-A_{I}}$$
where $A_{I}$ represents the intersection area, $A_{1}$ refers to the area of the powdery mildew bounding box, and $A_{2}$ refers to the area of the leaf bounding box.

## 3. Results and Discussion

### 3.1. Experimental Platform

The experiments in this study were conducted on a server equipped with two NVIDIA GeForce RTX 3090 GPUs, each with 24 GB of memory. The server ran on the Ubuntu 22.04 operating system, with further details provided in Table 1. The models were trained for 300 epochs, with a batch size of 32. The optimizer used was SGD, starting with an initial learning rate of 0.01, which was adjusted to a final value of 0.1. A momentum of 0.937 was applied, and the weight decay was set to 0.0005.

### 3.2. Evaluation Metrics

In this study, we utilized precision (P), recall (R), mean average precision (mAP), F1-score, model parameters, and FLOPs to comprehensively assess the performance of the models.

Precision measures the model’s capability to correctly identify positive samples and is computed as follows: (15)$$P=\frac{TP}{TP+FP}$$

Recall reflects the model’s ability to identify all relevant positive samples and is calculated as follows: (16)$$R=\frac{TP}{TP+FN}$$

F1-score is the harmonic mean of precision and recall, balancing the two metrics. The calculation formula is as follows:(17)$$F1=2\cdot \frac{P\cdot R}{P+R}$$

mAP averages precision across various recall levels and is calculated as follows:(18)$$mAP=\frac{1}{C}\sum_{c=1}^{C}AP_{c}$$

The parameter count reflects the model’s complexity, indicating the number of trainable parameters.

FLOPs measure the computational cost and are calculated as follows:(19)$$\mathrm{FLOPs}=2\times H_{\mathrm{out}}\times W_{\mathrm{out}}\times C_{\mathrm{out}}\times \left(K^{2}\times C_{\mathrm{in}}\right)$$
where TP is the number of true positive samples; FP is the number of false positive samples; *P* is precision; $F\bar{N}$ is the number of false negative samples; P(r) is the precision as a function of recall r;C is the total number of classes; and $AP_{c}$ is the average precision for class $c.\;H_{\mathrm{out}},\;W_{\mathrm{out}},\;C_{\mathrm{out}},\;K$, and $C_{\mathrm{in}}$ represent the output dimensions, number of output channels, kernel size, and input channels, respectively.

### 3.3. Detection Results of PM-YOLO

By utilizing the rubber tree powdery mildew dataset during the training, testing, and validation phases of the PM-YOLO model, we achieved efficient detection and precise localization of powdery mildew, as illustrated in Figure 9. Notably, the model’s performance in detecting small-target powdery mildew was significantly enhanced, enabling the accurate identification of diseased areas against complex backgrounds. This advancement provided an efficient tool for rubber tree disease management and served as a valuable reference for the automatic detection of diseases in other crops.

### 3.4. Comparison to Other Classical Models

To validate the performance of various models in the visual detection of rubber tree powdery mildew, this study selected several classic deep-learning-based detection algorithms for experimentation. Table 2 presents the experimental results for Faster R-CNN, SSD, YOLOv5, YOLOv8, YOLOv9, RT-DETR, and YOLOv10. Compared to one-stage networks Faster R-CNN and SSD, PM-YOLO achieved a 7.6% and 12.7% increment in mAP and a 2.9% and 1.5% increment in recall, respectively. The PM-YOLO model outperformed the classic YOLO and RT-DETR algorithms in both mAP and recall while maintaining only 466.9 M parameters and 28.8 G FLOPs. Furthermore, our proposed method improved mAP by 4.4% over YOLOv5, 5.6% over YOLOv8, 5.1% over YOLOv9, and 7.6% over YOLOv10 at the same parameter level. Additionally, PM-YOLO demonstrated significant improvements in precision and F1-score compared to the baseline YOLOv10-N while maintaining low FLOPs. It also exhibited a clear advantage in computational complexity over RT-DETR, with FLOPs as low as 28.8 G.

These experimental results indicated that, by integrating the FFDM module, which leverages multi-scale feature fusion, and the DASI module for enhancing the detection of small targets, PM-YOLO demonstrated exceptional performance in the detection of rubber tree powdery mildew. It is particularly suitable for scenarios requiring high precision and computational efficiency, thus making it an attractive choice for practical applications.

### 3.5. Ablation Experiments

To evaluate the effectiveness of PM-YOLO in detecting rubber tree powdery mildew, ablation experiments were conducted by incrementally adding the FDM, FFDM, and DASI to the base model. The results, shown in Table 3, demonstrate that these enhancements in model architecture and algorithmic strategies were effective. The addition of the FDM improved the propagation and fusion of the features, the FFDM enhanced the multi-scale feature extraction, and the DASI module significantly boosted the detection of small targets. Compared to the original YOLOv10 model, PM-YOLO achieved a 7.6% increment in mAP and an 8.2% increment in recall while also demonstrating greater efficiency in parameter count and computational load compared to the other state-of-the-art models.

Additionally, compared to the other advanced object detection models, PM-YOLO maintained high accuracy while significantly reducing both the number of parameters and computational requirements, making it more suitable for real-time and resource-constrained application scenarios. These ablation experiments not only validate the effectiveness of PM-YOLO but also offer valuable insights for the design of future object detection models.

## 4. Discussion

The proposed PM-YOLO framework offers an innovative solution for the precise and real-time detection and grading of powdery mildew on rubber trees, addressing critical challenges in agricultural disease management. By processing multi-scale inputs and capturing rich contextual information through parallel deep convolutions, the framework enhances feature extraction, particularly for small targets, enabling the timely identification of subtle pathological changes and facilitating early disease intervention. The integrated automated grading algorithm further improves the accuracy and reliability of disease assessment, reducing manual errors and supporting informed decision-making. Supported by a robust dataset of 6200 images and 38,000 annotations collected across different growth stages, PM-YOLO demonstrates strong generalizability under diverse environmental conditions. This model significantly advances multi-scale feature extraction, small-target detection, and disease management strategies, paving the way for future developments in precision agriculture.

Furthermore, in this study, we applied heatmap visualization techniques to better visualize which regions of an image contributed significantly to the classification. Specifically, we employed advanced EigenCAM [26] and GradCAM++ [27] methods, as shown in Figure 10. These methods improved the interpretability of the detection results and offer better insight into the model’s decisions. For instance, EigenCAM efficiently displayed the importance of feature maps for specific category outputs, while GradCAM++ enhanced the sensitivity to subtle changes, aiding in the identification of fine disease features.

The application of these visualization techniques showed that PM-YOLO accurately captured the characteristics of powdery mildew, enhancing both the model’s detection accuracy and the transparency of its predictions. This supported its use for the early detection and precise management of the disease.

However, future research should address some limitations. Although the dataset included various environments and disease levels, expanding it to cover different seasons and more complex environments will improve the model’s generalization. Additionally, challenges such as occluded leaves or poor lighting conditions can be mitigated by integrating multimodal data, such as hyperspectral and RGB images. Extending PM-YOLO’s application to other crops can further demonstrate its potential in agricultural disease management.

## 5. Conclusions

In conclusion, this study introduced the PM-YOLO model for the automatic detection and grading of powdery mildew on rubber tree leaves. By integrating the FFDM and DASI, the model significantly improved the detection accuracy of small targets in complex backgrounds. PM-YOLO surpassed the current state-of-the-art detection methods in both accuracy and recall while maintaining low computational costs. Additionally, the automatic grading algorithm provided a reliable tool for early disease management, reducing the dependence on subjective human judgment. Despite these achievements, future research should focus on expanding the dataset to cover more environmental conditions and enhancing the model’s robustness in real-world applications. Overall, PM-YOLO offered a promising solution for precision agriculture and disease management, with broad potential for application across various crops.

## Figures and Tables

**Figure 1 insects-15-00937-f001:**
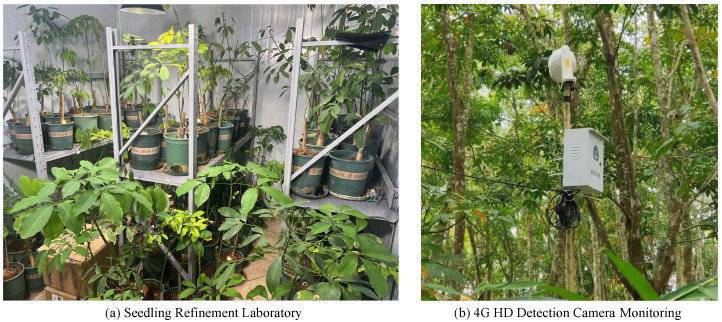
The Seedling Refinement Laboratory of Sanya Institute of Breeding and Multiplication (**a**) provides optimal growing conditions for rubber trees. The 4G HD Detection Camera Monitoring installed at Jinjiang Farm (**b**) enables real-time disease monitoring and prevention in the field. Both facilities support the research on rubber tree diseases.

**Figure 2 insects-15-00937-f002:**
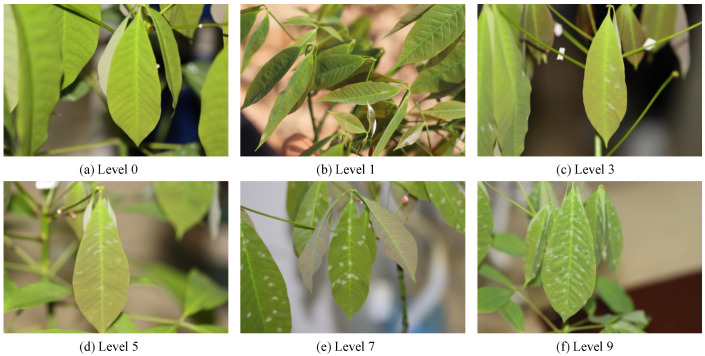
The dataset’s images of rubber tree powdery mildew depict varying disease severity from healthy (Level 0) to severely infected (Level 9), offering a visual guide to study the disease’s effect on leaf health.

**Figure 3 insects-15-00937-f003:**
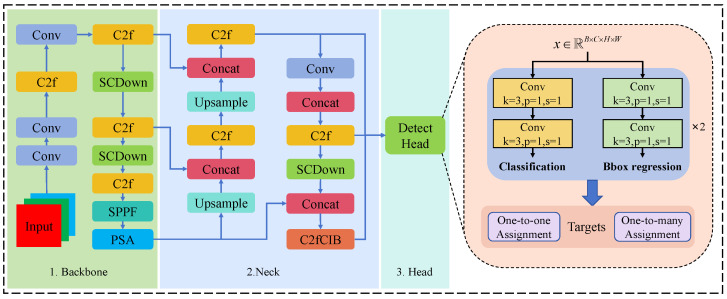
YOLOv10 model consisted of three main components: backbone, neck, and head. The head incorporated convolutional layers for classification and bounding box regression, utilizing both one-to-one and one-to-many assignment strategies to refine object detection accuracy.

**Figure 4 insects-15-00937-f004:**
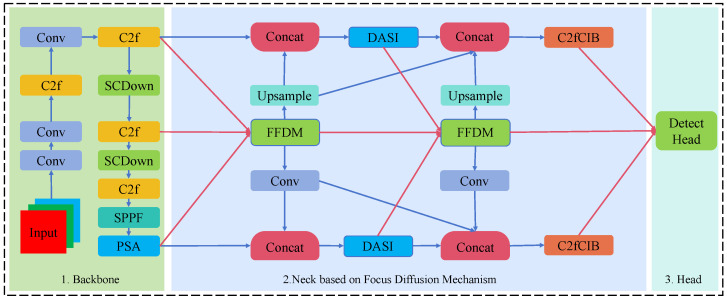
PM-YOLO model. The red and blue arrows in the figure represent the focusing and diffusing processes, respectively. By concentrating on features at different hierarchical levels and employing the FFDM module for multi-scale feature integration, contextually enriched features are effectively propagated across various detection scales.

**Figure 5 insects-15-00937-f005:**
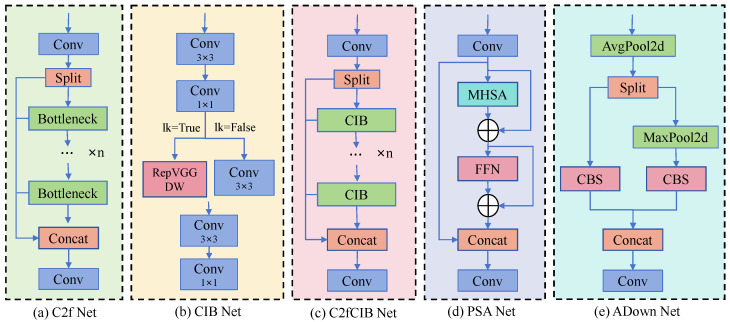
PM-YOLO incorporates five key modules: the C2f block, CIB block, C2fCIB block, PSA block, and ADown block. These modules each employ different methodologies to effectively handle complex image processing tasks.

**Figure 6 insects-15-00937-f006:**
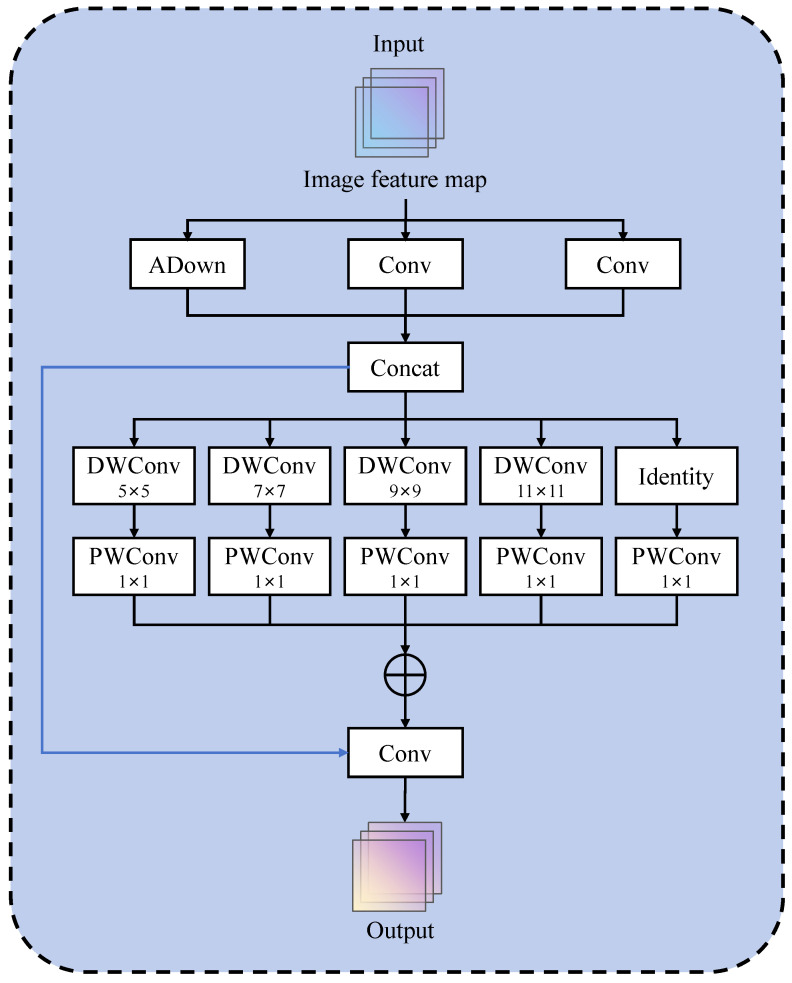
FFDM module.

**Figure 7 insects-15-00937-f007:**
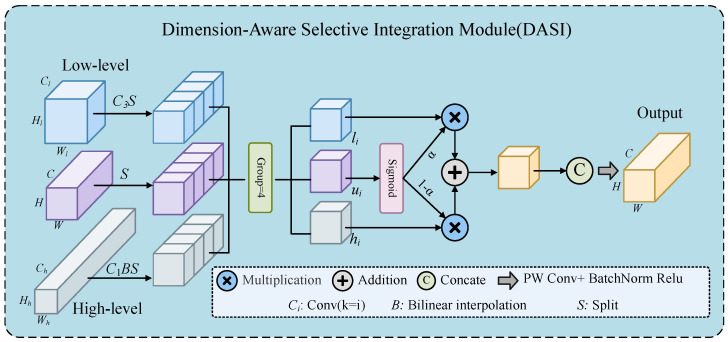
DASI module.

**Figure 8 insects-15-00937-f008:**
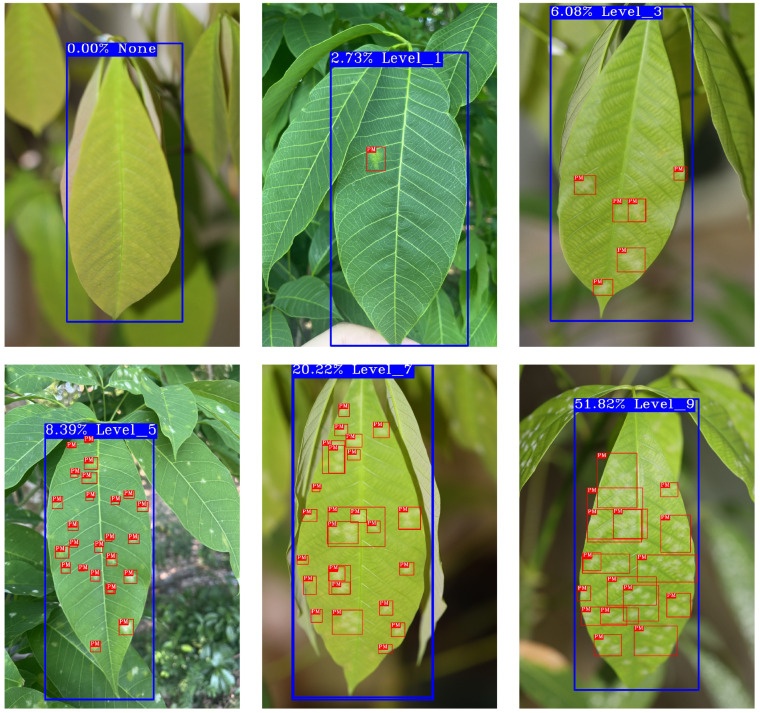
Automatic grading detection of powdery mildew.

**Figure 9 insects-15-00937-f009:**
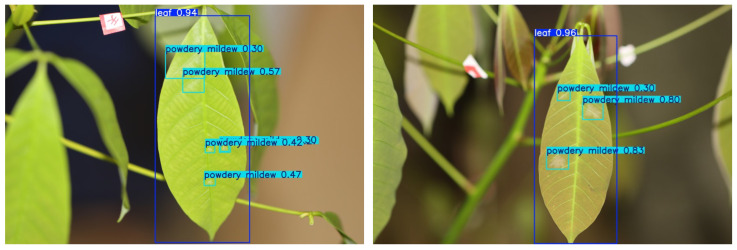
Detection results of PM-YOLO.

**Figure 10 insects-15-00937-f010:**
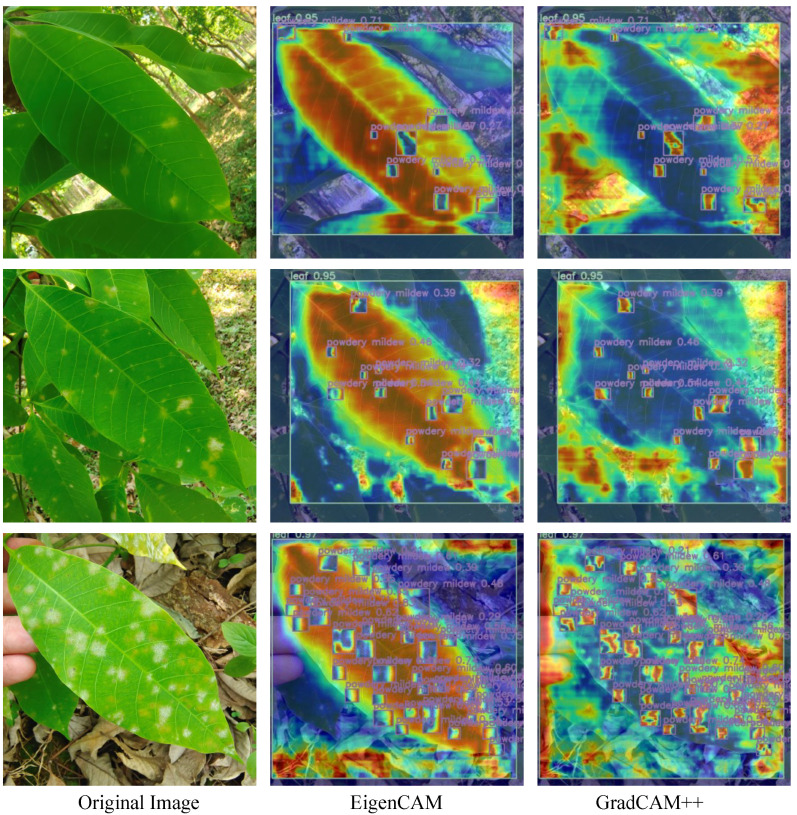
Thermal map visualization of the results.

**Table 1 insects-15-00937-t001:** Details of the experimental platform.

Hardware Platform or Software Environment	Model Identity or Designation	Parametric or Version
CPU	Intel Xeon Platinum 8474C	Frequency: 3.00 GHz
GPU	NVIDIA GeForce RTX 3090	Video memory: 24 GB (×2)
Operating System	Ubuntu	20.04
Deep learning framework	PyTorch	2.2.2
Computational platform	CUDA	12.1
Programming language	Python	3.9

**Table 2 insects-15-00937-t002:** Detection results of comparison experiment on rubber tree powdery mildew dataset.

Model	mAP/%	Precision/%	Recall/%	F1-Score/%	Param/M	FLOPs/G
Faster R-CNN	79.3	77.9	76.3	78.9	306.2	246.4
SSD	74.2	79.3	84.8	82.0	242.1	64.2
YOLOv5-N	81.6	84.5	78.5	81.2	263.1	8.2
YOLOv8-N	81.3	87.2	82.4	84.9	250.9	7.2
YOLOv9-C	81.8	81.1	79.4	80.0	456.5	232.9
YOLOv9-E	82.5	82.2	80.5	81.0	694.0	244.9
RT-DETR-R18	80.4	86.3	82.8	85.3	2018.4	58.6
RT-DETR-R34	78.7	79.3	81.4	79.9	3144.1	90.6
RT-DETR-R50	81.6	79.6	79.9	80.8	4294.4	134.8
RT-DETR-L	81.9	82.6	81.6	82.1	3297.0	108.3
YOLOv10-N	79.3	78.9	77.4	78.0	270.7	8.4
YOLOv10-S	79.2	78.0	79.5	78.6	806.8	40.6
YOLOv10-M	80.8	80.3	81.4	81.3	1648.7	64.0
YOLOv10-L	81.9	82.6	79.9	80.8	2576.8	127.2
YOLOv10-B	83.6	84.3	78.9	82.1	2045.4	98.7
YOLOv10-X	80.8	79.9	79.8	80.0	3165.8	171.0
**PM-YOLO**	**86.9**	84.8	**85.6**	85.2	466.9	28.8

**Table 3 insects-15-00937-t003:** Detection results of ablation experiments on rubber tree powdery mildew dataset.

Tag	Basic Model	+FDM	+FFDM	+DASI	mAP	Precision	Recall	F1-Score	Param	FLOPs
1	✓				79.3%	78.9%	77.4%	78.0%	270.7 M	8.4 G
2	✓	✓			81.9%	80.7%	80.6%	81.4%	320.3 M	12.4 G
3	✓	✓	✓		84.1%	82.1%	82.4%	83.4%	427.3 M	22.7 G
4	✓	✓	✓	✓	**86.9%**	**84.8%**	**85.6%**	**85.2%**	466.9 M	28.8 G

## Data Availability

Data will be made available on request.

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
