# Peer review of "PM-YOLO: A Powdery Mildew Automatic Grading Detection Model for Rubber Tree"

_insects, 2024, doi:10.3390/insects15120937_

Round 1

Reviewer 1 Report

Comments and Suggestions for Authors

This study deals with a novel deep learning-based approach for detecting powdery mildew in rubber trees which can help in early detection and intervention of the disease. The study is interesting and present some important results, however, some modifications should be performed before the manuscript can be acceptable. The manuscript needs to be revised for English language. Many corrections and comments are included in the attached file. 

Comments on the Quality of English Language

The manuscript needs to be deeply revised for English language. 

Author Response

Comment: This study deals with a novel deep learning-based approach for detecting powdery mildew in rubber trees which can help in early detection and intervention of the disease. The study is interesting and present some important results, however, some modifications should be performed before the manuscript can be acceptable. The manuscript needs to be revised for English language. Many corrections and comments are included in the attached file. 

Respond: 

Thank you very much for your valuable feedback and thoughtful comments on our manuscript. We have carefully reviewed the attached file and addressed all the issues and corrections you raised. Specifically, we have:

  1. Revised the manuscript for English language to enhance clarity and readability.
  2. Incorporated the modifications and suggestions provided in the attachment, ensuring that all identified errors and concerns have been thoroughly addressed.

We believe these changes have significantly improved the quality of the manuscript, and we hope that it now meets the standards for publication. Please let us know if there are any further concerns or suggestions.

Thank you again for your constructive feedback.

Reviewer 2 Report

Comments and Suggestions for Authors

The authors have done commendable work in addressing the significant issue of powdery mildew in rubber trees, which notably affects yield and quality. Their approach, modifying  YOLO  to PM-YOLO for early detection, showcases an effective use of modern machine learning techniques in agricultural disease management.

Overall, the manuscript is also well-structured, and the methodology is sound.

Although I would recommend that this article can be accepted in its present form, I would like the authors to consider the following minor suggestions to improve manuscript

  1. References: To enhance the article's utility and accessibility, I recommend that the authors include DOI URLs in the references. This addition would assist readers in locating cited works more efficiently.
  2. Results Presentation: Including frames per second (FPS) in the results section would provide a clearer understanding of the model's real-time performance. This metric is crucial for evaluating the practical applicability of the proposed system in field conditions.
  3. 4G Camera Model Inference Processing Location: The authors not that a live 4G enabled camera was used for real-time detection of powdery mildew disease, it would be great if the authors note how the model inference was done, including if the images where streamed to a remote server for inference or that if inference was done locally on the 4G camera enabled device.

Overall, this article is a valuable contribution to this journal and agricultural technology research, and the suggested adjustments would further strengthen its impact.

Author Response

Comments 1:References: To enhance the article's utility and accessibility, I recommend that the authors include DOI URLs in the references. This addition would assist readers in locating cited works more efficiently. 

Response 1:Thank you for your suggestion. We have addressed this issue by including DOI URLs for references in the manuscript to enhance accessibility and assist readers in locating the cited works more efficiently.

Comments 2:Results Presentation: Including frames per second (FPS) in the results section would provide a clearer understanding of the model's real-time performance. This metric is crucial for evaluating the practical applicability of the proposed system in field conditions. 

Response 2:In our work, "real-time" refers to monitoring within relatively short time intervals, such as half a day or several hours, rather than continuous live video streaming. To achieve this, we utilized fixed-interval static photo captures instead of video. Therefore, FPS is not a necessary evaluation metric for our system. This approach ensures effective disease monitoring while managing data processing and transmission efficiently.

Comments 3:4G Camera Model Inference Processing Location: The authors not that a live 4G enabled camera was used for real-time detection of powdery mildew disease, it would be great if the authors note how the model inference was done, including if the images where streamed to a remote server for inference or that if inference was done locally on the 4G camera enabled device.

Response 3:Thank you for the insightful comment. In our study, we used a Hikvision DS-NVR-F104/P/4G camera for remote monitoring. The camera is configured to automatically rotate and adjust focus to capture images from predefined angles at half-day intervals. These images are streamed to the front-end for monitoring personnel to conduct real-time inspections. We have also added relevant details about this process in the revised manuscript to provide a clearer understanding of our methodology.

Please let us know if there are any further concerns or suggestions. Thank you again for your constructive feedback.

Round 2

Reviewer 1 Report

Comments and Suggestions for Authors

All corrections were applied, however there few minor corrections need to be applied before acceptance as indicated in the attached file. 
